# Mediating role of food insecurity in the relationship between perceived MSM related stigma and depressive symptoms among men who have sex with men in Nepal

Kiran Paudel [1,2], Prashamsa Bhandari [3], Kamal Gautam[2], Jeffrey A. Wickersham[4], Toan Ha[5], Swagata Banik[6], Roman Shrestha [1,2,4]*

1 Nepal Health Frontiers, Tokha-5, Kathmandu, Nepal, 2 Department of Allied Health Sciences, University of Connecticut, Storrs, CT, United States of America, 3 Institute of Medicine, Tribhuvan University, Kathmandu, Nepal, 4 Department of Internal Medicine, Section of Infectious Diseases, Yale School of Medicine, New Haven, CT, United States of America, 5 School of Public Health, University of Pittsburgh, Pittsburgh, PA, United States of America, 6 Baldwin Wallace University, Berea, OH, United States of America

* roman.shrestha@uconn.edu

**Data Availability Statement:** The raw data in used for the analysis of this study have been deposited

## Abstract

### Background

Previous studies have indicated the association between stigma and depressive symptoms among frequently stigmatized groups, such as men who have sex with men (MSM). While this association has been suggested in the literature, there is a dearth of evidence that examines whether food insecurity statistically mediates the relationship between stigma and depressive symptoms.

### Methods

This cross-sectional study conducted between October and December 2022 among a sample of 250 MSM in Kathmandu, Nepal, recruited through respondent-driven sampling. An unadjusted model including the exposure (stigma), mediator (food insecurity), and outcome variables (depressive symptoms) and an adjusted model that controlled for sociodemographic, behavioral, and health-related confounders were used. Bootstrapping was utilized to estimate the coefficients of these effects and the corresponding 95% confidence intervals. Via bootstrap approach, we find out the mediating role of food insecurity in the association between stigma and depressive symptoms.

### Results

Depressive symptoms, stigma, and food insecurity were 19.6%, 24.4%, and 29.2%, respectively, among MSM. Food insecurity was associated with higher age (b = 0.094; 95% CI = 0.039, 0.150) and monthly income (b = -1.806; 95% CI = -2.622, -0.985). Depressive symptoms were associated with condom-less sex in past six months (b = -1.638; 95% CI = -3.041, -0.092). Stigma was associated with higher age (b = 0.196; 95% CI = 0.084, 0.323) and PrEP uptake (b = 2.905; 95% CI = 0.659. 5.248). Food insecurity statistically mediated 20.6% of the indirect effect of stigma in depressive symptoms.

in the public data repository platform "Fig share" and can be easily accessed using the link 10.6084/m9.figshare.23750913.

**Funding:** This work was supported by a career development grant from the National Institute on Drug Abuse [K01DA051346] for RS. The funders had no role in study design, data collection and analysis, decision to publish, or preparation of the manuscript.

**Competing interests:** The authors have declared that no competing interests exist.

## Conclusion

Our findings show that food insecurity affects the relationship between stigma and depressive symptoms in this population. Reducing food insecurity and addressing the stigma surrounding sexual orientation should be a priority when addressing mental health concerns among MSM in Nepal and other resource-limited countries with similar socio-cultural settings.

## Introduction

The detrimental effects of stigma on an individual's mental health have been well documented. Stigma is a social process that systematically devalues someone, or a group of people based on their perceived differences from the dominant group. Studies have shown that sexual minorities (including men who have sex with men; MSM) are particularly exposed to stigma, lack of social support, minority stress, social and legal discrimination, identity concealment, internalized homophobia, and abuse, specifically based on their sexual orientation, increasing their risk of psychological disorders [1]. Lifetime and current prevalence rates of depression among MSM are far higher than those of their heterosexual counterparts [2, 3]. The unique stressors, such as prejudice and sexual-minority-specific internalized stressors, among that particular population [4], can partially explain why MSM are more prone to the risk of mental health concerns, such as depression.

Available evidence suggests that MSM are among the vulnerable groups that are likely to experience higher rates of food insecurity due to historical patterns of minority stress and stigma, which may increase avoidance of social interactions [5, 6]. For MSM, hostile and stressful social conditions are theorized to deplete social and economic resources, including employment opportunities, wages, and social connections, resulting in unequal distribution of health risks, including heightened vulnerability to food insecurity [7]. While some studies have shown an association between food insecurity and mental health, including depression [8], it is imperative to broaden the theoretical lens by considering additional dimensions. For instance, the intricate association between minority stress, stigma, and food insecurity suggests that the stigma associated with food insecurity can create a feedback loop, increasing the vulnerability of marginalized groups, including MSM [9]. Moreover, stigma not only accompanies but is exacerbated by food insecurity, potentially intensifying the negative impact on mental health outcomes [9].

Recognizing and addressing modifiable contextual factors, particularly food insecurity, is crucial for developing comprehensive strategies aimed at reducing stigma and depression among MSM [6, 9]. The interaction between stigma, food insecurity and depression among MSM is complex. By recognizing this complex interaction, our study aims to postulate a pathway outlining the relationship between these variables. Therefore, we tried to postulate a pathway outlining the relationship among these variables. Specifically, we hypothesize that the relationship between stigma and depression is mediated by food insecurity. This could enable us to explore the potential pathways, providing an improved understanding of the development of depression among MSM. Furthermore, this nuanced understanding could help in understanding the predictors of depression and designing efficacious interventions aimed at preventing depression in MSM.

Nepal remains unexplored with respect to the concerns of food insecurity, stigma, and depression studied together among MSM. Although Nepal's legislation is considered

progressive towards gender and sexual minorities, they seem to have to face negative attitudes and exclusion from their societies and families. These social prejudices stemmed from the traditional patriarchal culture, including forced marriages of MSM to females can create an environment in aggravating the situation of sexual stigma [10]. The social and economic marginalization of sexual and gender minorities in Nepal, can also contribute to MSM experiencing food insecurity [11]. These conditions then eventually can lead the pathway to depression.

To the best of our knowledge there are no estimates of the association between food insecurity and depressive symptoms among MSM in low resource setting country Therefore, in the present study, we investigated the factors affecting depression among MSM in Nepal. Given the association between stigma and depression among MSM, as shown in previous studies, we focused our analysis on whether this association is mediated through food insecurity.

## Methods

### Study design and participants

This cross-sectional study was conducted among MSM who were at least 18 years' old, understood Nepali or English, were willing to undergo HIV and Syphilis screening and residing in the Kathmandu Valley of Nepal. The Kathmandu Valley is comprised of three districts: Kathmandu, Bhaktapur, and Lalitpur. Kathmandu district is the national capital, densely populated, and the largest metropolitan city whereas Bhaktapur and Lalitpur are neighboring districts inside the valley.

### Recruitment and procedures

In our study, 250 MSM were enrolled by using Response Driven Sampling (RDS) method from October to December 2022. RDS have been found effective for recruiting hidden and hard-to-reach population [12]. We purposively selected five MSM "seeds" based on recommendations from community-based organizations led by LGBTIQ and gave them 5 recruitment coupons to distribute to potential participants. Each subsequent participant was also given 5 recruitment coupons to recruit additional peers [13]. Since we took a survey with participants who were voluntarily interested in taking part and eligible for the study, we did not have any non-response rate.

Participants were compensated 1000 Nepalese Rupees (equivalent to approximately USD 8) for participating in the study. In addition, for each of up to five eligible peers they successfully recruited into the study, they received an extra incentive of 500 Nepalese Rupees (~ USD 4).

### Measures

The study utilized a questionnaire that assessed various characteristics of MSM, including food insecurity, MSM related stigma and depressive symptoms. The questionnaire was first developed in English and then translated into Nepali with the assistance of experts. Before being used in the study, it was pretested on five MSM who were not included in the final study sample. The Cronbach's alpha coefficient was used to evaluate the internal consistency of the translated questionnaire.

### Participants' characteristics

The collected data of participants included the following characteristics: age, level of education (up to grade 10, and high school and above), monthly income, sexual orientation, disclosure of sexual orientation, relationship status, Pre-exposure Prophylaxis (PrEP) usage, and

condomless sex in the past six months. Sexual orientation was classified as gay (cis gender male who engaged in sexual activities with man only) and bisexual (cis gender male who engaged in sexual activities with male and female).

## Food insecurity

The widely used and practical instrument by the United Nations eight item Food Insecurity Experience Scale (FIES) was used to measure food insecurity [14]. The questionnaire focused on self-reported behaviors and experiences related to food access and resource constraints in the past 12 months. The Food Insecurity Experience Scale (FIES) was used to measure the presence and severity of food insecurity, with scores ranging from 0 (no symptoms of food insecurity) to 8 (all symptoms of food insecurity). FIES scores was obtained adding the items to obtain aggregated score and score greater than 3 were used to determine food insecurity (moderate to severe food insecurity), as suggested previously, whereas in the mediational model, the FIES continuous scale was used [14]. The reliability of the scale in our sample was 0.94, as measured by Cronbach's alpha, which is high.

## Stigma

Participant's perceived MSM related stigma was assessed using a 14-items scale based on Neilands Sexual Stigma Scale [15], which was modified by Logie et al. [16]. Responses to the 14 questions were measured on a 4-point Likert scale and were scored from 0–3 giving a total stigma score of 0–42. A score greater than 10 was considered stigmatized. Stigma was represented by a summative score; higher scores indicating greater stigma [15, 16]. The reliability of the scale in our sample was 0.82 as measured by Cronbach's alpha.

## Depressive symptoms

Depressive symptoms were assessed using the Patient Health Questionnaire-9 (PHQ-9), a nine-item scale that measures the frequency of symptoms experienced over the past two weeks. Each item was scored on a scale from 0 (not at all) to 3 (almost every day), and the total score ranged from 0 to 27, with higher scores indicating more frequent and severe symptoms [17]. For the mediation analysis, the PHQ-9 scores were used in a continuous form. The Cronbach's alpha coefficient for the PHQ-9 in this study population was 0.85.

## Statistical analysis

Data were analyzed using SPSS V.26. Descriptive statistics were used to summarize the characteristics of the sample, including frequencies and percentages for categorical variables and means, standard deviations, and ranges for continuous variables. To identify the factors associated with food insecurity, stigma, and depressive symptoms, we conducted multiple linear regression analyses with all relevant variables and no variable was found to have a higher variance inflation factor (VIF) score greater than 5.

To assess the robustness of these models, we employed bootstrapping with 5000 replications. We used the PROCESS macro, which is based on ordinary least squares regression, to test the hypothesized mediating effects of food insecurity on stigma and depressive symptoms. We ran two models: an unadjusted model including the exposure, mediator, and outcome variables and an adjusted model that controlled for sociodemographic, behavioral, and health-related confounders. Bootstrapping was used to estimate the coefficients of these effects and the corresponding 95% confidence intervals. A non-parametric bootstrap approach, which is not based on the assumption of normal distribution was used.

## Results

### General characteristics of the participants

This study included 250 MSM with a mean age of 27.6±8.9 years old. More than half of the participants studied at least plus two/higher secondary (58.0%) and had an income of more than Nepalese rupee twenty thousand (54.8%). Nearly half of the participants (48.4%) reported having had condomless sex, and only around one-third (30.4%) had used pre-exposure prophylaxis in the past six months. Few of the respondents did not disclose their sexual orientation to anyone (14.8%) and nearly two third of them identified themselves as gay (63.2%) (**Table 1**).

### Prevalence of food insecurity, stigma, and depressive symptoms, and their correlates

Among the study participants, the prevalence of depressive symptoms, stigma and food insecurity were 19.6%, 24.4%, and 29.2% respectively is presented in **Fig 1**. MSM with at least two of the three conditions constituted 15.2% and those with all three conditions constituted 4.4%. as shown in **Fig 2**.

The results of the multivariate linear regression analyses examining the relationship between MSMs' characteristics and food insecurity, MSM related stigma, and depressive symptoms are presented in **Table 2**. Overall, MSM's older age was associated with food insecurity (b = 0.094; BCa 95% CI = 0.039, 0.150) and participants with monthly income of 20, 000 and above was associated with food insecurity (b = -1.806; 95% CI = -2.622, -0.985).

**Table 1. Characteristics of the participants (N = 250).**

| Variables | | Number | Percentage |
|---|---|---:|---:|
| Age | | | |
| | Mean ± SD | 27.6 ± 8.9 | Min:18; Max:70 |
| Level of education | | | |
| | Up to grade 10 | 105 | 42 |
| | High school and above | 145 | 58 |
| Income level | | | |
| | Less than NRs 20000 (USD ~150) | 113 | 45.2 |
| | NRs 20000 and above | 137 | 54.8 |
| Sexual orientation | | | |
| | Gay | 158 | 63.2 |
| | Bisexual | 92 | 36.8 |
| Relationship status | | | |
| | Single | 161 | 64.4 |
| | With partner | 89 | 35.6 |
| Never had condomless sex in the past six months | | | |
| | Yes | 94 | 51.6 |
| | No | 88 | 48.4 |
| Ever used of PrEP | | | |
| | Yes | 76 | 30.4 |
| | No | 174 | 69.6 |
| Disclosed sexual orientation to anyone | | | |
| | Yes | 213 | 85.2 |
| | No | 37 | 14.8 |

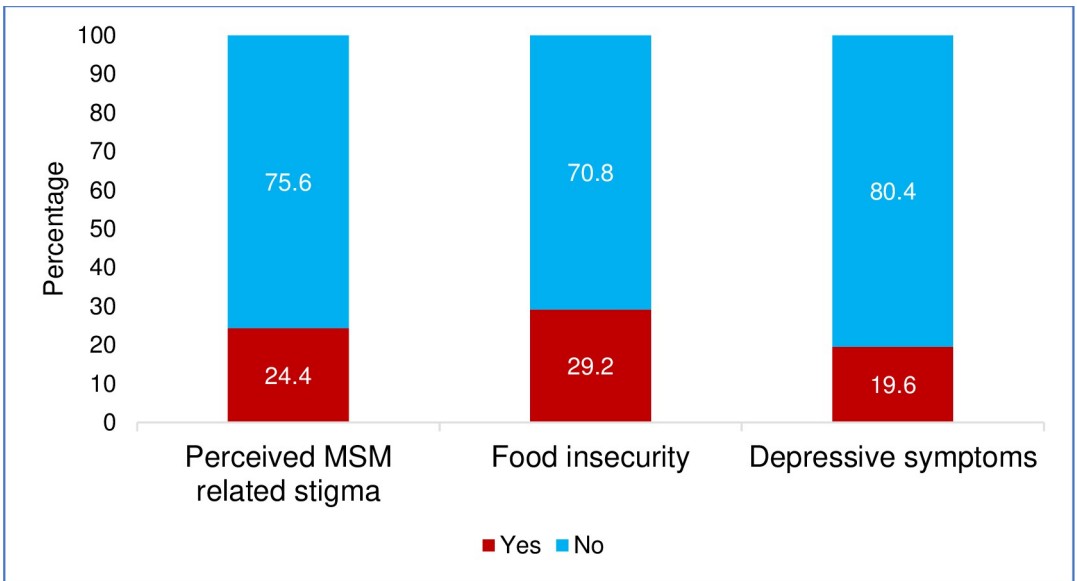

**Fig 1. Prevalence of depressive symptoms, food insecurity, and perceived MSM-related stigma.**

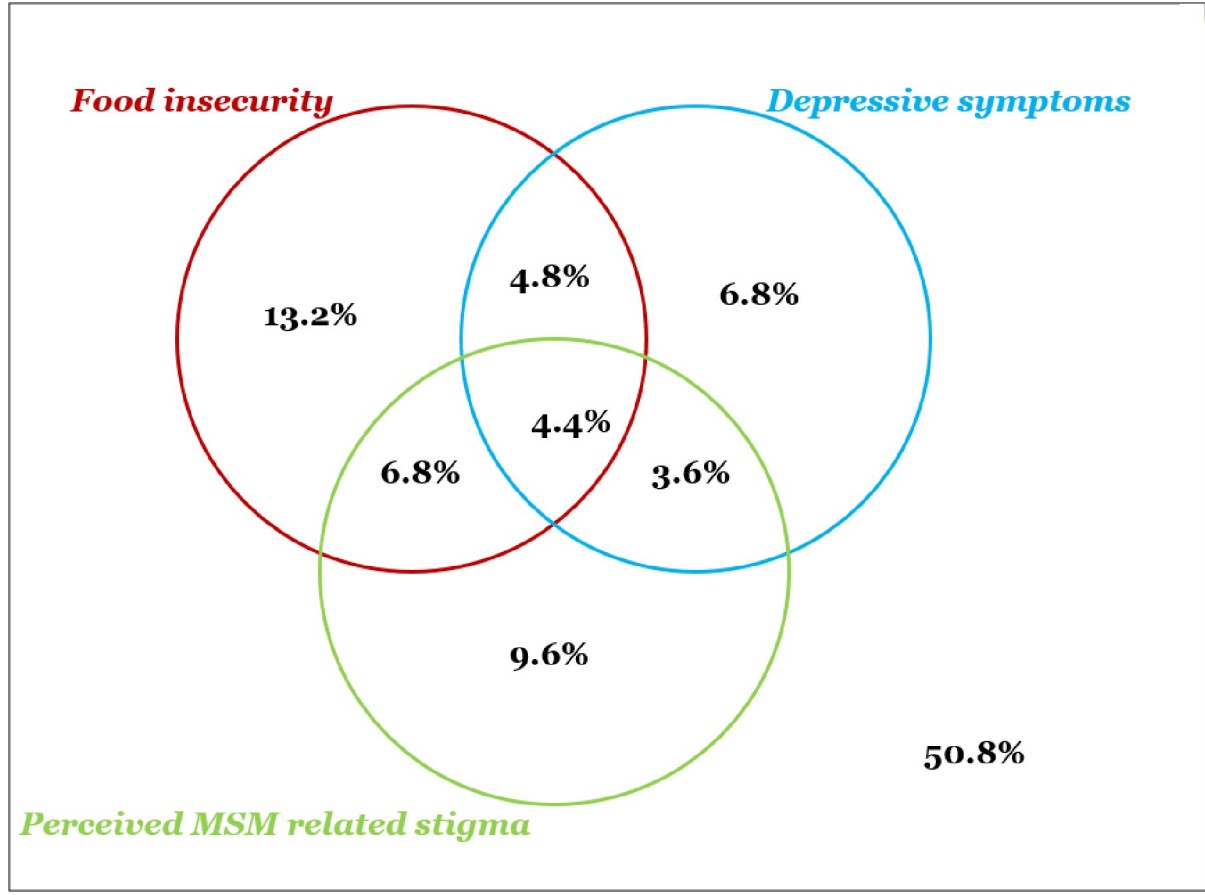

**Fig 2. Venn diagram depicting depressive symptoms, food insecurity and perceived MSM related stigma.**

**Table 2. Multivariable linear regression models of MSM characteristics with food insecurity, stigma, and depressive symptoms (N = 250).**

| Characteristics | Food insecurity | | Stigma | | Depressive symptoms | |
|---|---|---|---|---|---|---|
| | B | Bca 95% CI | b | Bca 95% CI | B | Bca 95% CI |
| Age | 0.094 | 0.039, 0.156 | 0.196 | 0.084, 0.323 | 0.049 | -0.33, 0.131 |
| Level of education (Ref: Up to SLC) | -0.348 | -1.290, 0.570 | 0.221 | -2.198, 2.531 | 1.676 | -0.010, 3.413 |
| Income (Ref Less than 20,000) | -1.806 | -2.622, -0.985 | -1.905 | -3.924, 0.106 | -1.437 | -3.089,0.146 |
| Sexual orientation (Ref: Gay) | 0.153 | -0.770, 1.133 | -2.767 | -4.538, -1.051 | -1.413 | -2.844,0.064 |
| Relationship status (Ref: Single) | -0.416 | -1.294, 0.476 | -0.430 | -2.601. 1.819 | -1.504 | -3.099, 0.116 |
| Disclosed to anyone (Ref: No) | 0.571 | -0.851, 2.078 | -4.346 | -6.300, -2.526 | -2.398 | -3.740, -1.002 |
| Ever used of PrEP (Ref; No) | 0.677 | -0.127, 1.492 | 2.905 | 0.659. 5.248 | 0.399 | -1.176, 1.902 |
| Condomless sex in the last six months (Ref: Yes) | -0.202 | -0.990, 0.628 | -1.295 | -3.103, 0.696 | -1.638 | -3.041, -0.092 |

Participants who always used condoms during sex in the last six months (b = -1.638 95% CI = -3.041, -0.092) and who disclosed their sexual orientation (b = -2.398; 95% CI = -3.740, -1.002) had lower depressive symptoms than their counterparts. Higher age was associated with higher stigma in MSM (b = 0.196; 95% CI = 0.084, 0.323) and participants who ever used PrEP (b = 2.905; 95% CI = 0.659. 5.248) had a higher stigma. Participants who are gay had higher stigma than their bisexual counterparts (b = -2.767; 95% CI = -4.538, -1.051).

## Mediation of association between stigma and depressive symptoms by food insecurity

The results of the mediation analysis with food insecurity as the mediator showed that 20.6% of the relationship between stigma and depressive symptoms was explained by food insecurity. This finding was after considering sociodemographic and behavioral factors in model 2 as shown in **Table 3**.

## Discussion

Men who have sex with men have distinctive experiences regarding depressive symptoms and food insecurity compared to their heterosexual counterparts, such as systemic marginalization and stigma associated with being part of a marginalized community. In this study, food insecurity accounted for 20.6% of the effect of stigma on depressive symptoms. This suggests that depression in MSM is caused by a combination of factors, including food insecurity and stigma, rather than by either of these factors independently.

**Table 3. Effect estimates of effects of stigma on depressive symptoms among MSM mediated via food insecurity (N = 250).**

| Characteristics | Model 1 | | Model 2 | |
|---|---|---|---|---|
| | b (SE) | BCa 95% CI | b (SE) | BCa 95% CI |
| Indirect effect | 0.0535 (0.0199) | 0.0203, 0.0991 | 0.0564 (0.0231) | 0.183, 0.1089 |
| Direct effect | 0.2250 (0.0446) | 0.1371, 0.3128 | 0.2173 (0.0521) | 0.1145, 0.3200 |
| Total effect | 0.2785 (0.0439) | 0.1921, 0.3649 | 0.2737 (0.0510) | 0.1729, 0.3744 |
| The proportion of total effect mediated. | 0.1921 | | 0.206 | |
| The ratio of indirect to direct effect | 0.2377 | | 0.259 | |

1. Model 1: an unadjusted mediational model

2. Model 2: adjusted for sociodemographic + behavioral variables (age, education, income, sexual orientation, relationship status, disclosed to anyone, ever used of PrEP, condomless sex in last six months)

3. b: unstandardized coefficient; BCa, Bias corrected and accelerated: 5000 bootstrap samples

Our observed prevalence of food insecurity (29.2%) was higher than that of other studies reported among MSM [18] and other sexual minorities like transgender, female sex workers [9]. Evidently, food insecurity among Nepali at the national level is comparatively higher than that of other countries, because Nepal is predominantly reliant on agriculture and is facing a growing trade deficit [19]. The COVID-19 pandemic has only further exacerbated these issues; because of their status as a minority group, MSM are more vulnerable to food insecurity due to evident socioeconomic disparities and lack of access to social safety nets during such catastrophic times. Recent studies examining the effects of food insecurity beyond nutrition have noted that it contributes to an elevated risk of mental health problems. Food insecurity can lead to depression through multiple potential behavioral as well as biological channels [8], including stress caused by the struggle to secure food. Therefore, the study suggests that the relief support plans and policies should focus on the implementation of immediate sustainable food insecurity strategies to prevent food insecurity problems among MSM in Nepal, with a focus on addressing the structural stigma and discrimination that may consequently lead to food insecurity. On the other hand, decreasing food insecurity also requires addressing the determinants of economic instability among MSM. In our study, MSM with less income were significantly associated with food insecurity, which can be corroborated by other studies as well [11, 20]. Several studies have found high rates of unemployment and employment discrimination among sexual and gender minority individuals [21, 22], leading to greater levels of economic inequality. This lack of financial stability can lead to difficulties in purchasing food and maintaining food security, especially when resources are scarce.

As reported in this study, about one-fourth of MSM face stigma based on their sexual orientation. Nepal has provided legislative protection to sexual and gender minorities against discrimination and affirmed the fundamental rights of sexual and gender minorities in its constitution [23]. Despite the progressive legislation, MSM are still confronted with struggles in their daily lives, primarily because of the social stigma associated with their sexual orientation. They are subjected to discrimination, exclusion and ostracism within their families and society. They often feel pressured to conform to heteronormative standards, which discourages them from openly expressing their sexuality. Furthermore, social stigma is also a source of negative feelings, such as having to conceal one's true sexual identity from family and friends [10, 21]. Thus, internalized homophobia can cause individuals to internalize negative messages about their sexuality, resulting in lower self-esteem, increased feelings of shame, and negative self-image. These feelings of low self-worth can lead to depression [16, 24].

Additionally, resilience and social support, which can act as positive buffers between stress and health outcomes, are usually lacking for sexual minorities, resulting in feelings of isolation, which is associated with greater rates of depression. Our study reported a higher experience of stigma among gays than among bisexual male, which is in contrast to a study done in the US [6]. Moreover, PrEP utilization has been associated with increased engagement in high-risk sexual behaviors among MSM, leading to the perception of promiscuity among PrEP users [25]. Furthermore, MSM who use PrEP are likely to undergo more frequent HIV testing, which in turn may lead to increased experiences of stigma [26]. MSM who use PrEP are more likely to have HIV and it's evident that MSM with HIV/AIDS and who tested HIV frequently can feel more stigmatized [27].It was also seen that older MSM faced more stigma. Several studies have restated a commonly held belief that the gay community places more emphasis on youth [28], which might be the reason that experience of stigma is seen higher in older men who have sex with men in this study as well. Moreover, younger MSM may be more likely to come out at younger age, while older MSM may have struggled with their sexuality for longer periods and may have experienced more trauma or stigma and discrimination as a result.

Among the MSM studied in this research, 19.6% reported having depression, which was higher than the reported national prevalence of depression in Nepal (11.7%) [29]. MSM who did not use condoms during sex and those who did not disclose their sexual orientation were found to be more at risk of having depression. Misinformation about condoms and HIV has likely caused many MSM to not use them regularly, and to feel uncomfortable asking their partners to do so. This leads to lower self-efficacy, which is a factor leading to unprotected anal sex [30]. Unprotected sex can lead to the feelings of guilt, vulnerability, anxiety, and fear of contracting HIV and other STIs. Collectively, these factors may contribute to the increased risk of depression. Likewise, pressure to disclose one's identity as an MSM brings about strong internalized stigma, leading to increased psychological distress [31]. In addition, those who have not disclosed their identity may experience a lack of social support, and thus might be more likely to internalize negative messages and beliefs about their own sexual orientation, leading to feelings of guilt, shame, and self-loathing, which can consequently contribute to depression. Despite these reasons, this might have happened because of screening tools used by our study and the study of national prevalence of depression in Nepal.

There are reports of an association between of stigma and depression among MSM [16, 24]. However, the potential mechanism by which food insecurity mediates the relationship may be discussed for several reasons. First, stigma can lead to a sense of social isolation and exclusion, which can prevent individuals from accessing support networks that could otherwise provide them with food or assistance in accessing food resources [32]. In addition, it can also lead to a lack of self-esteem and internalized homophobia, which can restrict individuals from seeking opportunities and resources, leading to financial instability and subsequently, limiting their ability to access food [7]. Conversely, studies have reported the influence of food insecurity on the prevalence of depression among MSM [8]. Additionally, individuals who face food insecurity may also be more likely to engage in risky activities, such as sex work, in order to secure food and other essentials [33]. Stigma can be the source of distress that could lead to food insecurity and may turn in depressive symptoms [6, 34]. Stigma-induced food insecurity can hinder individuals from sustaining self-sufficiency by causing a decline in economic productivity or job loss, thereby contributing to, or exacerbating depressive symptoms [6, 34, 35]. To elucidate this relationship, qualitative, and longitudinal quantitative data are needed. Stigma surrounding a particular issue can serve as a source of distress, potentially causing a cascade of problems. To address these challenges effectively, it is important to implement distress coping strategies like social support, self-care, problem solving, and so on.

There are several limitations on the study that need to be addressed. First, the cross-sectional nature of the study design precludes the ability to establish causality or directionality in the relationship between these variables. Second, reliance on self-reported measures may have led to underestimation of the true extent of depression, stigma, and food insecurity among participants due to social desirability bias. Thirdly, the temporal association in this context may pose challenges because we assessed depressive symptoms over the past two weeks and food insecurity over the last twelve months. Fourthly, there is potential chance of reverse causality because individuals with depressive symptoms may be more inclined to perceive both food insecurity and stigma. Lastly, we recommend future researchers to do such study and analysis considering the family income and wealth quintile which might have significant impact on this study. To address these limitations, future studies should utilize qualitative and longitudinal data to gain a more nuanced understanding of the complex relationships between food insecurity, stigma, and depressive symptoms among MSM. Additionally, future studies should explore potential covariates that may impact the mediation models tested in this study. By gaining a comprehensive understanding of these relationships, we can develop more

effective interventions and policies to improve the health and well-being of MSM in low-resource settings, like Nepal.

## Conclusion

Our study shows that food insecurity mediates the association between stigma and depressive symptoms among MSM. The process of mediation, however, should be investigated in more detail in future studies that employ larger longitudinal designs and have a longer follow-up period across these population groups. And, considering the limited availability of mental health services and social support for this marginalized group, any future efforts to address mental health concerns among MSM in Nepal and other settings with limited resources, should prioritize reducing food insecurity and addressing the stigma surrounding their sexual orientation.

## Supporting information

**S1 File. Survey questionnaire.**
(PDF)

## Acknowledgments

The authors would like to acknowledge all the study participants involved in the study and the staff of Blue Diamond Society. We are also grateful towards Prashuram Chaudhari, and Manisha Dhakal for helping us during this study.

## Author Contributions

**Conceptualization:** Kamal Gautam, Roman Shrestha.

**Data curation:** Kiran Paudel, Kamal Gautam.

**Formal analysis:** Kiran Paudel, Roman Shrestha.

**Funding acquisition:** Roman Shrestha.

**Investigation:** Kamal Gautam, Roman Shrestha.

**Methodology:** Kiran Paudel, Prashamsa Bhandari, Kamal Gautam, Roman Shrestha.

**Project administration:** Kiran Paudel, Kamal Gautam, Roman Shrestha.

**Resources:** Kamal Gautam, Jeffrey A. Wickersham, Swagata Banik, Roman Shrestha.

**Software:** Kiran Paudel, Roman Shrestha.

**Supervision:** Kamal Gautam, Jeffrey A. Wickersham, Toan Ha, Swagata Banik, Roman Shrestha.

**Validation:** Toan Ha, Roman Shrestha.

**Visualization:** Kiran Paudel, Jeffrey A. Wickersham, Swagata Banik.

**Writing – original draft:** Kiran Paudel, Prashamsa Bhandari, Kamal Gautam.

**Writing – review & editing:** Kiran Paudel, Prashamsa Bhandari, Kamal Gautam, Jeffrey A. Wickersham, Toan Ha, Swagata Banik, Roman Shrestha.

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
