## [Decision Letter · Decision Letter 0]

16 Oct 2023

PONE-D-23-26721Mediating Role of Food Insecurity in the Relationship Between Sexual Minority Stigma and Depressive Symptoms Among Men Who Have Sex with Men in NepalPLOS ONE

Dear Dr. Shrestha,

Thank you for submitting your manuscript to PLOS ONE. After careful consideration, we feel that it has merit but does not fully meet PLOS ONE’s publication criteria as it currently stands. Therefore, we invite you to submit a revised version of the manuscript that addresses the points raised during the review process.

We look forward to receiving your revised manuscript.

Kind regards,

Cesar Infante Xibille, Ph.D

Academic Editor

PLOS ONE

Reviewers' comments:

Reviewer's Responses to Questions

**Comments to the Author**

1. Is the manuscript technically sound, and do the data support the conclusions?

Reviewer #1: Yes

Reviewer #2: Yes

Reviewer #3: Partly

2. Has the statistical analysis been performed appropriately and rigorously? 

Reviewer #1: Yes

Reviewer #2: I Don't Know

Reviewer #3: Yes

3. Have the authors made all data underlying the findings in their manuscript fully available?

Reviewer #1: Yes

Reviewer #2: Yes

Reviewer #3: Yes

4. Is the manuscript presented in an intelligible fashion and written in standard English?

Reviewer #1: No

Reviewer #2: Yes

Reviewer #3: Yes

5. Review Comments to the Author

Reviewer #1: Overall, I congratulate the authors on this interesting contribution to understanding food insecurity’s role as a mediating factor between stigma and depressive symptoms in MSM. The core arguments are clear and well-aligned, and the statistical tests are appropriate for the sample. The authors have done a nice job of presenting simple and easily understandable figures, though the readability of the tables could be improved. Though most language is good, I recommend an overall proofread before considering direct publication because there are a few errors which significantly influence interpretation. I recommend this article for publication after minor revisions. My detailed comments are in the attached document.

Reviewer #2: This article addresses and under-researched issue, the relationship between food insecurity and mental health in marginalized populations, and I think it contains valuable information.

My main recommendation is to develop the theoretical basis for the authors’ arguments. In the paragraph in p. 10 beginning “Available evidence suggests that MSM are among the vulnerable groups that are likely to experience higher rates of food insecurity...”, they reference one article about the stigma of FI itself (ref. 5), another about health disparities in general (ref. 7), and yet another on the relationship between FI and mental health (with no reference to stigma) (ref. 8). I suggest the authors expand on their argument, using refs. 6 & 9 and other similar ones (some of which they use in the Discussion already).

Maybe because of this limited theoretical background, in p. 11 the authors seem to suggest two alternative hypotheses (either FI  stigma DS, or stigma  FI  DS), before describing their aim as testing only one of those “In the present study, we investigated the factors affecting depression among MSM in Nepal. Given the association between stigma and depression among MSM, as shown in previous studies, we focused our analysis on whether this association is mediated through food insecurity.” (stigma  FI  DS). What made the authors decide in favor of the second mediation hypothesis?

Likewise, the authors don’t provide justification for the other variables (confounders?) included in the analysis. Given the argument of the article is about the relationship between stigma, FI and DS, what’s the relevance of the variables condomless sex and PrEP? And of the other variables added to the mediation model?

Other suggestions:

• As the authors measured perceived stigma, I suggest they reflect this in the title and other sections of the article (not “stigma” but “perceived stigma”)

• In the abstract: “... there is a dearth of evidence that examines whether food insecurity mediates this relationship and if stigma plays a statistically significant role in the association between food insecurity and depressive symptoms in MSM.” In this phrase the two hypotheses mentioned above are mixed. I suggest the authors revise the text to make sure they are consistent in their descriptions of the expected relationships.

• In the abstract: “Multiple linear regression analysis was used to examine the factor associated”. I suggest specifying they employed path analysis, since their aim was to test for mediation.

• A limitation that the authors don’t mention is the potential for reverse causality, with those having DS more likely to perceive both FI and stigma.

• Could the authors specify if they scored the FIES using the Rasch method proposed by FAO, or added the items to obtain and aggregated score? This is not clear from the description in p. 13

• The authors employed a method based on OLS to test for mediation. However, the scores in the PHQ are not typically distributed as a normal variable, and the association between PHQ scores and other variables are frequently non-linear. Did the authors test for the assumptions of OLS? Does PROCESS allow for using other distributions? Please clarify.

• Could the authors detail the rationale for using bootstrapping, the bootstrapping approach employed, and add references for these?

• Fig. 1 repeats the same information that is given in the last par. of p. 16. I don’t think the figure is needed.

• The Venn diagram is interesting, but it seems more suited to interaction than mediation. Could the authors justify its utility?

• In the Discussion: “In this study, food insecurity accounted for 20.6% mediation of the total effect of stigma on depressive symptoms” The correct interpretation is that food insecurity accounted for 20.6% of the effect of stigma on depressive symptoms

• In the Discussion, the prevalence of DS in this study is compared to that of a national study in Nepal (ref. 27). However, the national study employed a different scale for DS. The authors should mention this limitation.

• The authors mention as a limitation that “... reliance on selfreported measures may have led to underestimation of the true extent of depression, stigma, and food insecurity among participants due to social desirability bias”. These could happen in other studies as well. Why do the authors think this a limitation particularly of this study?

• They also state that “Lastly, the study did not collect data on other forms of food insecurity, such as poverty-related food insecurity,” How do they know their study did not measure “poverty-related food insecurity”? I don’t understand the authors’ point here

• I’m not a native English speaker, but I felt there were numerous language errors and maybe a few typos (e.g. “This cross-sectional study was conducted among MSM who 18 years were at least old”)

Reviewer #3: the authors present an interesting exploration regarding stigma and depressive symptoms and the possible mediating impact of food (in) security in nepali msms. the authors propose that the association between stigma and worsening depressive symptoms is mediated by food security/ insecurity. however, the authors do not credibly show that the effect seen is purely due to food security rather than a highly correlated / related exposure such as lower income.

the authors chose to use the FIES instrument but it is not clear that the population fulfilled the criteria for using this instrument (designed to examine population level insecurity, and not necessarily designed for this type of analysis.) it would be important to specify the distribution of the responses to the 8 FIES questions and the distribution of the scores using the FIES software in order to determine what proportion had moderate vs Severe food insecurity. (even if this will not effect the mediatrion analyses since those have been done with the continuous scores.

the authors may want to consider using the FIES indicators classifying moderate or severe food insecurity rather than the continous score.

it would also be important to consider temporality and to consider whether the FIES was applied regarding last month's diet, or whether it was applied regarding the prior 12 months. Similarly, it would be important to know if the time frame for the PHQ9 and stigma scales and how these compare to the time period addressed in the FIES.

a minor point: the authors might want to be careful when using the term 'better food insecurity' as at times perhaps they mean lesser food security. it would be helpful to check this carefully throughout.

it is not clear why the authors did not more closely examine whether mediation between stigma and depression was not because of lower income, rather than because of food insecurity, especially since these are highly correlated. one would expect that other factors may also be potential mediators such as housing situation.

THere is no discussion of including Housing situation in the models. This would be an important consideration.

overall this is an important contribution and I hope these additional details can be added.

6. PLOS authors have the option to publish the peer review history of their article (what does this mean?). If published, this will include your full peer review and any attached files.

Reviewer #1: No

Reviewer #2: No

Reviewer #3: No

---

## [Author Response · Author response to Decision Letter 0]

30 Nov 2023

Reviewer #1: 

1. Overall, I congratulate the authors on this interesting contribution to understanding food insecurity’s role as a mediating factor between stigma and depressive symptoms in MSM. The core arguments are clear and well-aligned, and the statistical tests are appropriate for the sample. The authors have done a nice job of presenting simple and easily understandable figures, though the readability of the tables could be improved. Though most language is good, I recommend an overall proofread before considering direct publication because there are a few errors which significantly influence interpretation. I recommend this article for publication after minor revisions. My detailed comments are below:

Response: We appreciate your positive response. Thank you. 

2. Please check style throughout to avoid incomplete statements, mixed word order, and apparent typos which impact understanding, e.g., in Abstract, “The impact of food stigma on depressive symptoms […]” e.g. in Measures, “[…] educational status (you mean level?), monthly income […]” e.g. in Statistical analysis, “[…] mediating effects of medication on these outcomes.”

Response: Thank you for flagging these issues. We have now addressed them in the revised manuscript. 

3. Mention a few key details about the FIES as used, namely 1) was the reference period 30 days or 12 months?, 2) given the sample size and type, clarify to what extent unadjusted raw score was used vs. applying infit/outfit/etc. measures?, and 3) not necessary everywhere, but here in measures please specify that the concept of “food insecurity” in this paper appears to be equivalent with “moderate to severe food insecurity” according to the FIES

Response: Thank you for the suggestion. The reference period of FIES was ‘in past 12 months’. We would also like to confirm that the food insecurity here indicates ‘moderate to severe food security’ according to FIES. We have now made some edits within the document to make it clear. 

4. Please indicate if any cut-off values were used to classify stigma, and if so, how these relate or were adapted from previous literature.

Response: Thank you for this comment. In our study, we opted not to use a cut-off value for stigma, treating it as a continuous variable in both multivariable and mediation analysis. However, we did dichotomize in Figure 1, where a score greater than 10 was considered as stigmatized. This information has been included in the revised manuscript.

5. “Participants who are gay had higher stigma than their bisexual counterparts.” This is the first time the terms “gay” and “bisexual” are used. If you want to use them, please define them before the Results section by linking them clearly to the term “MSM”

Response: Thank you for your suggestion. We have addressed this definition for the terms "gay" and "bisexual" in the methods section of the manuscript. 

6. Given the potentially sensitive nature of these questions, I suggest noting levels of non-response. Related to that, you could indicate n in Table 2, to keep the general sample size in mind but also note by item (if relevant) any higher non-response to certain items as compared to others.

Response: Thank you for your feedback. We've updated Table 2 to note the sample size and a non-response rate. We have now clarified this in the Methods section that we did not encounter any non-responses as all eligible participants enrolled in the study.

7. It’s not immediately clear why the bolded values are bolded. Make a footnote or explicitly address each one in the text.

Response: We appreciate the reviewer's observation regarding the bolded values. To address the potential confusion, we have removed the bold formatting from all variables to ensure a consistent presentation. Thank you for bringing this to our attention.

8. Clarify if the result stated in the text comes from Model 1 or Model 2. Unless I’m misunderstanding, Table 3 shows the that 20.6% comes from Model 2, but this seems contradicted by the text just before the table.

Response: Thank you for your observation. The reported result, 20.6%, is indeed from Model 2. We've clarified this in the manuscript for accuracy. Appreciate your feedback.

9. I find myself wondering about the “socio-demographic” variable of income being used as an adjustment variable. Income would be very strongly linked with food insecurity, especially as measured by the FIES. How did the authors consider this relationship in the analysis and discussion?

Response: Thank you for your insightful question about using income as an adjustment variable. To address this, we want to clarify that we specifically measured individual monthly income rather than family income. This choice was intentional, as we aimed to focus on individual economic factors. We understand the importance of this consideration, especially for participants under 25 years, many of whom were students and financially dependent on their families. We recognize the potential impact of family income on the results. We have acknowledged this in the limitations section and suggested future researchers consider family income. 

10. “Our observed prevalence of food insecurity (29.2%) was higher than that of other studies reported among MSM (16) and other sexual minorities like transgender, female sex workers (9). Evidently, food insecurity among Nepali at the national level is comparatively higher than that of other countries, because Nepal is predominantly reliant on agriculture and is facing a growing trade deficit (17). I’d suggest caution when making conclusions about this which make assumptions about the entire population, as you do in the second sentence. Have you considered how your sampling strategy as compared with other studies may have affected these values?

Response: Thank you for pointing out the potential issue. Given the absence of specific studies on food insecurity among MSM in Nepal, we aimed to emphasize the broader concern in the general population. To address the limitation, we've added a note in the discussion section stating, "Despite these reasons, this might have happened because of screening tools used by our study and the study of the national prevalence of depression in Nepal."

11. Broadly, the Discussion would benefit from consideration of depression as a potential contributor to food insecurity and stigma, as opposed to only exploring the other direction. Throughout the introduction, modeling, and limitations the multi-directionality of these pathways is considered, but in my view it’s not well-addressed in the discussion. 

Response: Thank you for your constructive feedback. We have carefully incorporated the suggested changes. We believe these changes help in enhancing the overall quality of manuscript. 

12. “Additionally, individuals who face food insecurity may also be more likely to engage in risky activities, such as sex work, in order to secure food and other essentials (31).” Suggest incorporating the term “coping strategies” to describe actions like these, to benefit the authors in future literature searches around this developing topic.

Response: Thank you for your valuable input. We appreciate your suggestion and have now incorporated the term coping strategies into the discussion as “Stigma surrounding a particular issue can serve as a source of distress, potentially causing cascade of problems. To address these challenges effectively, it is important to implement distress coping strategies like social support, self-care, problem solving, and so on.”

13. “Lastly, the study did not collect data on other forms of food insecurity, such as poverty-related food insecurity, which limits our ability to fully understand the role of food insecurity in the relationship between stigma and depression.” I disagree with this statement as I understand it. The FIES questions explicitly explore “poverty-related” food insecurity (most food insecurity could be described that way). I’d remove this statement

Response: Thank you for your careful observation. By considering your suggestion we have now removed the confusing statement from the manuscript. 

Reviewer #2: 

 1. This article addresses and under-researched issue, the relationship between food insecurity and mental health in marginalized populations, and I think it contains valuable information.

My main recommendation is to develop the theoretical basis for the authors’ arguments. In the paragraph in p. 10 beginning “Available evidence suggests that MSM are among the vulnerable groups that are likely to experience higher rates of food insecurity...”, they reference one article about the stigma of FI itself (ref. 5), another about health disparities in general (ref. 7), and yet another on the relationship between FI and mental health (with no reference to stigma) (ref. 8). I suggest the authors expand on their argument, using refs. 6 & 9 and other similar ones (some of which they use in the Discussion already).

Response: Thank you for your constructive suggestion for enhancing the quality of the manuscript. We are pleased to inform you that we have incorporated the theoretical foundation into the introduction section of the manuscript as per your suggestion. 

2. Maybe because of this limited theoretical background, in p. 11 the authors seem to suggest two alternative hypotheses (either FI  stigma DS, or stigma  FI  DS), before describing their aim as testing only one of those “In the present study, we investigated the factors affecting depression among MSM in Nepal. Given the association between stigma and depression among MSM, as shown in previous studies, we focused our analysis on whether this association is mediated through food insecurity.” (stigma  FI  DS). What made the authors decide in favor of the second mediation hypothesis?

Response: Thank you for your valuable input. Now we revised the sentence and made align with our aim and removed the potentially confusing phrase from the paragraph.

3. Likewise, the authors don’t provide justification for the other variables (confounders?) included in the analysis. Given the argument of the article is about the relationship between stigma, FI and DS, what’s the relevance of the variables condomless sex and PrEP? And of the other variables added to the mediation model?

Response: Thank you for flagging the potential issues. We added these variables because the primary objective of the study was to evaluate HIV prevalence, sexually transmitted diseases, and mental health among MSM who may be at risk for HIV. Therefore, we identified these variables as potential confounders during the literature review. Consequently, we incorporated adjustments for these variables in the mediation model following all the statistical measures. 

4. Other suggestions:

• As the authors measured perceived stigma, I suggest they reflect this in the title and other sections of the article (not “stigma” but “perceived stigma”)

Response: Thank you for the suggestion. We have made necessary edits in the title and other sections of the manuscript, as suggested by the reviewer.

5. In the abstract: “... there is a dearth of evidence that examines whether food insecurity mediates this relationship and if stigma plays a statistically significant role in the association between food insecurity and depressive symptoms in MSM.” In this phrase the two hypotheses mentioned above are mixed. I suggest the authors revise the text to make sure they are consistent in their descriptions of the expected relationships.

Response: Thank you for your careful observation and valuable input. We have made the necessary changes to the abstract.

6. In the abstract: “Multiple linear regression analysis was used to examine the factor associated”. I suggest specifying they employed path analysis, since their aim was to test for mediation.

Response: Thank you for bringing up this point, we made changes in the abstract by removing multiple linear regression analysis and elucidating the primary aim of the study. 

7. A limitation that the authors don’t mention is the potential for reverse causality, with those having DS more likely to perceive both FI and stigma.

Response: Thank you for your valuable comment. It is crucial to consider the potential for reverse causality in our study, as individuals with depressive symptoms may be more inclined to perceive both food insecurity and stigma. Therefore, we have now added this as one of the limitations. 

8. Could the authors specify if they scored the FIES using the Rasch method proposed by FAO, or added the items to obtain and aggregated score? This is not clear from the description in p. 13

Response: We appreciate your valuable comment for making FIES aggregated score clearer. We have added items to obtain and aggregated score, now we mentioned this in the manuscript. 

9. The authors employed a method based on OLS to test for mediation. However, the scores in the PHQ are not typically distributed as a normal variable, and the association between PHQ scores and other variables are frequently non-linear. Did the authors test for the assumptions of OLS? Does PROCESS allow for using other distributions? Please clarify. Could the authors detail the rationale for using bootstrapping, the bootstrapping approach employed, and add references for these?

Response: Thank you for your concern. Considering your suggestion, we made changes on the manuscript statistical analysis section. However, for the best test of mediation effect, a non-parametric bootstrap approach, which is not based on the assumption of normal distribution, was used. Here are some of the articles which have followed the same statistical analysis as ours even if the variables are different nature of study and aim is similar. 

Bhandari, P.M., Neupane, D., Rijal, S. et al. Sleep quality, internet addiction and depressive symptoms among undergraduate students in Nepal. BMC Psychiatry 17, 106 (2017). https://doi.org/10.1186/s12888-017-1275-5

Li X, Yan H, Wang W, Yang H, Li S. Association between enacted stigma, internalized stigma, resilience, and depressive symptoms among young men who have sex with men in China: a moderated mediation model analysis. Ann Epidemiol. 2021 Apr;56:1-8. doi: 10.1016/j.annepidem.2021.01.001. Epub 2021 Jan 7. PMID: 33422600.

Whitley DM, Kelley SJ, Lamis DA. Depression, Social Support, and Mental Health: A Longitudinal Mediation Analysis in African American Custodial Grandmothers. The International Journal of Aging and Human Development. 2016;82(2-3):166-187. doi:10.1177/0091415015626550 

10. Fig. 1 repeats the same information that is given in the last par. of p. 16. I don’t think the figure is needed.

Response: Thank you for your concern. The figure on page 16 provides detailed information allowing users to extract information directly from the figure rather than relying solely on the accompanying text. However, we would be happy to remove it from the manuscript if you or the editor think otherwise.

11. The Venn diagram is interesting, but it seems more suited to interaction than mediation. Could the authors justify its utility?

Response: We agree. The benefit of including the Venn diagram includes allowing the reader to visualize the overlap (and potential interaction) between three variables. We would be happy to remove it from the manuscript if you or the editor think otherwise.

12. In the Discussion: “In this study, food insecurity accounted for 20.6% mediation of the total effect of stigma on depressive symptoms” The correct interpretation is that food insecurity accounted for 20.6% of the effect of stigma on depressive symptoms

Response: Thank you for your keen observation now we corrected it in the manuscript. 

13. In the Discussion, the prevalence of DS in this study is compared to that of a national study in Nepal (ref. 27). However, the national study employed a different scale for DS. The authors should mention this limitation.

Response: Thank you for your careful observation. We have now incorporated your suggestion and written in manuscript as “Despite these reasons, this might have happened because of screening tools used by our study and the study of national prevalence of depression in Nepal.”

14. The authors mention as a limitation that “... reliance on self reported measures may have led to underestimation of the true extent of depression, stigma, and food insecurity among participants due to social desirability bias”. These could happen in other studies as well. Why do the authors think this a limitation particularly of this study?

Response: Thank you for bringing up important concerns. This is a kind of common and general limitation of most of the studies of this nature. However, MSM being more stigmatized in Nepalese context, there might be more chance of happening, so we stressed on it. 

15. They also state that “Lastly, the study did not collect data on other forms of food insecurity, such as poverty-related food insecurity,” How do they know their study did not measure “poverty-related food insecurity”? I don’t understand the authors’ point here

Response: Thank you for flagging this potential confusion among readers. As per suggestion from #reviewer 1, we deleted this confusing statement from the manuscript. 

16. I’m not a native English speaker, but I felt there were numerous language errors and maybe a few typos (e.g. “This cross-sectional study was conducted among MSM who 18 years were at least old”)

Response: Thank you for your careful observation. We made some corrections on the typos.

Reviewer #3: 

1.The authors present an interesting exploration regarding stigma and depressive symptoms and the possible mediating impact of food (in) security in Nepali msms. The authors propose that the association between stigma and worsening depressive symptoms is mediated by food security/ insecurity. however, the authors do not credibly show that the effect seen is purely due to food security rather than a highly correlated / related exposure such as lower income.

Response: We acknowledge the importance of considering alternative explanations for the observed effects, and we would like to provide additional clarification regarding our study design and data collection. To address this, we want to clarify that we specifically measured individual monthly income rather than family income. This choice was intentional, as we aimed to focus on individual economic factors. We understand the importance of this consideration, especially for participants under 25 years, many of whom were students and financially dependent on their families. We recognize the potential impact of family income on the results. We have acknowledged this in the limitations section and suggested future researchers consider family income. 

2.The authors chose to use the FIES instrument but it is not clear that the population fulfilled the criteria for using this instrument (designed to examine population level insecurity, and not necessarily designed for this type of analysis.) it would be important to specify the distribution of the responses to the 8 FIES questions and the distribution of the scores using the FIES software in order to determine what proportion had moderate vs Severe food insecurity. (even if this will not effect the mediation analyses since those have been done with the continuous scores. The authors may want to consider using the FIES indicators classifying moderate or severe food insecurity rather than the continuous score.

Response: We appreciate your thoughtful suggestion. We acknowledge the importance of the analysis requested by reviewer, as you rightly pointed out however, it falls beyond the intended scope of our study. Nevertheless, to provide clarity for our readers, we have already assessed the prevalence of food insecurity. We have taken it into careful consideration, and in response, we have made some changes to the method section. The FIES tool has been employed in the general population, and similar analyses have been conducted, as given below.

Wiss, D., Javanbakht, M., Li, M., Prelip, M., Bolan, R., Shoptaw, S., & Gorbach, P. (2021). Food insecurity partially mediates the association between drug use and depressive symptoms among men who have sex with men in Los Angeles, California. Public Health Nutrition, 24(13), 3977-3985. doi:10.1017/S1368980021002494

Hamill MM, Hu F, Adebajo S, Kokogho A, Tiamiyu AB, Parker ZF, Charurat ME, Ake JA, Baral SD, Nowak RG, Crowell TA; TRUST/RV368 Study Group. Food and Water Insecurity in Sexual and Gender Minority Groups Living With HIV in Lagos, Nigeria. J Acquir Immune Defic Syndr. 2023 Jun 1;93(2):171-180. doi: 10.1097/QAI.0000000000003183. PMID: 36881816; PMCID: PMC10293107.

Pandey, S., Fusaro, V. Food insecurity among women of reproductive age in Nepal: prevalence and correlates. BMC Public Health 20, 175 (2020). https://doi.org/10.1186/s12889-020-8298-4

Singh DR, Sunuwar DR, Shah SK, Sah LK, Karki K, Sah RK (2021) Food insecurity during COVID-19 pandemic: A genuine concern for people from disadvantaged community and low-income families in Province 2 of Nepal. PLoS ONE 16(7): e0254954. https://doi.org/10.1371/journal.pone.0254954

3. It would also be important to consider temporality and to consider whether the FIES was applied regarding last month's diet, or whether it was applied regarding the prior 12 months. Similarly, it would be important to know if the time frame for the PHQ9 and stigma scales and how these compare to the time period addressed in the FIES.

Response: Thank you for highlighting this concern. We have carefully considered your feedback and incorporated into the limitation section of our manuscript as “Thirdly, the temporal association in this context may pose challenges because we assessed depressive symptoms over the past two weeks and food insecurity over the last twelve months.”

4. A minor point: the authors might want to be careful when using the term 'better food insecurity' as at times perhaps they mean lesser food security. it would be helpful to check this carefully throughout.

Response: Thank you for bringing attention to this matter. We have now addressed your valuable input in our manuscript and incorporated necessary changes. 

5. It is not clear why the authors did not more closely examine whether mediation between stigma and depression was not because of lower income, rather than because of food insecurity, especially since these are highly correlated. one would expect that other factors may also be potential mediators such as housing situation. There is no discussion of including Housing situation in the models. This would be an important consideration. Overall, this is an important contribution and I hope these additional details can be added

Response: Thank you for your critical comment for making this manuscript better. We acknowledge the importance of considering alternative explanations for the observed effects, and we would like to provide additional clarification regarding our study design and data collection. In our study, we specifically measured the monthly income of the participants rather than that of their families. This deliberate choice was made to focus on individual economic factors that may not directly impact food insecurity because half of the participant were age less than 25 years and student and they were dependent one family so, this might not impact on the result. 

However, this is an important concern, so now we kept this in limitation section of manuscript as “Lastly, we recommend future researchers to do such study and analysis considering the family income and wealth quintile which might have significant impact on this study.”

---

## [Editor Report · Decision Letter 1]

6 Dec 2023

Mediating Role of Food Insecurity in the Relationship Between Perceived MSM Related Stigma and Depressive Symptoms Among Men Who Have Sex with Men in Nepal

PONE-D-23-26721R1

Dear Dr. Roman Shrestha,

We’re pleased to inform you that your manuscript has been judged scientifically suitable for publication and will be formally accepted for publication once it meets all outstanding technical requirements.

Kind regards,

Cesar Infante Xibille, Ph.D

Academic Editor

PLOS ONE
---

## [Editor Report · Acceptance letter]

20 Dec 2023

PONE-D-23-26721R1 

PLOS ONE

Dear Dr. Shrestha, 

I'm pleased to inform you that your manuscript has been deemed suitable for publication in PLOS ONE. Congratulations! Your manuscript is now being handed over to our production team.

Kind regards, 

on behalf of

Dr. Cesar Infante Xibille 

Academic Editor

PLOS ONE